# Study on Preparation of Superhydrophobic Surface by Selective Laser Melting and Corrosion Resistance

**Lei Xing** *,†, **Jingui Yu** †, **Zhiyong Ji, Xingjiu Huang, Chaoyuan Dai and Qiaoxin Zhang** *

School of Mechanical and Electronic Engineering, Wuhan University of Technology, 122 Luoshi Road,
Wuhan 430070, China; yujingui@whut.edu.cn (J.Y.); jizhiyong1027@163.com (Z.J.); xingjiuhuang@iim.ac.cn (X.H.);
dai18639818916@163.com (C.D.)

* Correspondence: xing435324368@163.com (L.X.); zhangqx@whut.edu.cn (Q.Z.)

† Authors contribute equally to this work.

**Abstract:** Superhydrophobic surfaces are used in aerospace, medical equipment, transportation, household appliances and other fields due to their special interface characteristics. In this paper, a superhydrophobic surface is prepared by Selective Laser Melting (SLM) 3D-printed technology, comparing the effects of different post-treatment methods and time on corrosion resistance, and revealing the root cause of the transition from hydrophilic to superhydrophobic. The test results show that for samples not treated with fluoro-silane, the microstructure adsorbs the organic matter in the air and reduces the surface energy, which is the root cause of the sample surface changing from hydrophilic to superhydrophobic. In addition, the corrosion resistance of 3D-printed, polished, 3D-printed + modified, and 3D-printed + corroded samples are analyzed. Among them, 3D-printed + modified samples have a longer resistance to corrosion, and after placing in outdoor natural conditions for 60 days, the contact angle of water droplets on the surface is 150.8°, which still has superhydrophobic properties and excellent natural durability.

**Keywords:** 3D printed; post-treatment process; superhydrophobic; wettability; corrosion resistance

## 1. Introduction

Superhydrophobic surface refers to the surface of a material that exhibits a water-repellent function under the combined action of micro–nano composite structures and special surface substances, and the contact angle with water droplets is greater than 150° [1–3]. In nature, many animal and plant surfaces have excellent superhydrophobic properties, such as lotus leaves, rose petals, water strider feet, beetle epidermis, butterfly wings, etc. [4–7] Due to the development of bionics, the phenomenon of superhydrophobicity has received widespread attention and has gradually become a research hotspot [8–11]. On the one hand, people continue to explore the relationship between the microstructure and the wetting characteristics of superhydrophobic surfaces and establish basic wetting models [12–14]; on the other hand, with the continuous deepening of research, researchers are no longer limited to the pursuit of high hydrophobicity of superhydrophobic surfaces, and they hope to achieve low-cost preparation and efficacy persistence of superhydrophobic surfaces by constructing specific wetting models and obtaining new preparation methods [15]. Based on the theoretical analysis of superhydrophobic models (Wenzel model and Cassie–Baxter model), the preparation of superhydrophobic surfaces can be achieved by surface roughening of low surface energy objects or modification of rough surfaces with low surface energy [16].

Currently, many different methods are employed to fabricate superhydrophobic surfaces, including machining [17], replica molding [18], sol-gel process [19], Chemical Vapor Deposition (CVD) [20], chemical etching [21], hydrothermal [22], etc. Esmaeilirad et al. used NaOH to remove the oxide layer on the aluminum surface and etched a conical-like microstructure, and then used an etching solution mixed with hydrochloric acid and acetic

acid to etch micro-pores on the surface microstructure, so that the sample surface had superhydrophobic performance [23]. Berendjchi et al. used the sol-gel method to obtain silica sol doped with different amounts of copper nanoparticles, and then dipped it on a cotton fabric substrate, dried and solidified, and finally used Hexadecyl Trimethoxysilane (HDTMS) modified to obtain a superhydrophobic surface with self-cleaning and antibacterial properties [24]. The rise of laser processing technology has enriched the processing methods of superhydrophobic surfaces. Wang et al.'s study based on the microscopic morphology of the fish body and the arrangement of the scales. Under the action of the laser beams, by controlling the density of the laser path and processing time, biomimetic fish-scale surfaces with an inclined surface was obtained on the aluminum surface. The wettability results showed that the sample with 100% zoom ratio demonstrated excellent superhydrophobicity [25].

Selective Laser Melting (SLM) 3D-printed technology is one of the laser processing methods; its equipment adopts multi-beam printing, and the printing accuracy is at the micron level, so it has the advantages of high efficiency, good printing accuracy and a controllable shape, which is suitable for surface topography processing [26,27]. This paper introduces Selective Laser Melting (SLM) 3D-printed technology into the preparation of superhydrophobic surfaces. In the early stages, the minimum size of the cylindrical structure prepared by the printer was roughly determined according to the printed accuracy and experimental exploration: the diameter of the cylinder $d \geq$ 150 μm, cylindrical clearance $s - d \geq$ 100 μm, where $s$ is the center distance between adjacent cylinders, the height of the cylinder $H \geq$ 10 μm. Tests show that when the cylinder diameter $d$ is 180 μm, the adjacent cylinder gap $s - d$ is 280 μm, and the cylinder height $H$ is 400 μm, the contact angle of the 3D-printed + modified sample is 153.4°. It meets the superhydrophobic conditions and has good superhydrophobicity. The aim of the work is to innovatively use 3D printing to prepare superhydrophobic surfaces, compare the effects of different post-treatment methods and time on corrosion resistance, and reveal the root cause of the surface transition from hydrophilic to superhydrophobic.

## 2. Experiments

### 2.1. Sample Preparation

The preparation of samples are as follows:

(1) The preparation steps of 3D-printed and 3D-printed + modified samples are as follows: firstly, according to the design criteria of the superhydrophobic cylindrical array structure, use 3D Solidworks software to design the cylindrical array model with a cylinder diameter $d$ of 180 μm, gap between adjacent cylinders $s - d$ of 280 μm, and the cylinder height H of 400μm, and export the STL format. Secondly, import the STL model into the slicing software (Magics Print Metal for HanBang). Then, use an HBD-80 metal 3D printer (Hanbang Laser Technology Co., Ltd., Zhongshan, China) to print and obtain 3D-printed samples (laser power of 160 W, scanning speed of 1000 mm/s, scanning layer thickness of 30 μm, the scanning strategy is strip scanning, the shielding gas is argon, and the printing material is 250 mesh 316L stainless steel powder (Hanbang Laser Technology Co., Ltd., Zhongshan, China)). The above process is shown in Figure 1. Then, soak the sample in the anhydrous alcohol solution of fluoro-silane (Aladdin Reagent Co., Ltd., Shanghai, China). Take out the sample after soaking for 12 h, and finally put it into a DHG-9075A constant temperature blast drying oven (Yiheng Scientific Instrument Co., Ltd., Shanghai, China) at 90 °C for 2 h to obtain 3D-printed + modified samples. Fluoro-silane easily forms hydrogen bonds with the hydroxyl groups on the surface of the sample, which greatly improves the hydrophobicity of the surface and prevents direct contact between the surface and the outside world.

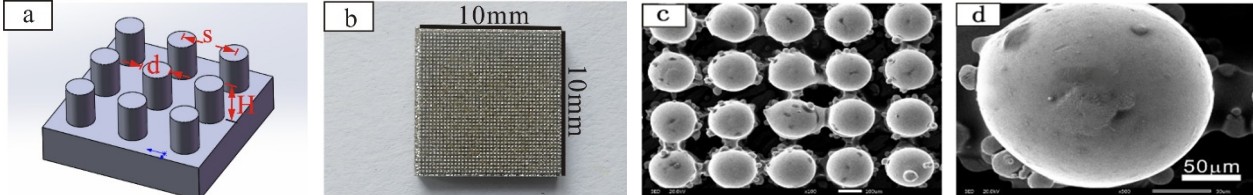

**Figure 1.** The cylindrical array structure. (**a**) Model; (**b**) printed entity; (**c**) SEM image magnified 100 times; (**d**) SEM image magnified 500 times.

(2) The preparation steps of 3D-printed + corroded samples are as follows: Firstly, use Solidworks software to establish the cylindrical array microstructure model, and the characteristic dimensions of the cylindrical microstructure are the same as above. Slice the established 3D model and set the parameters as above. Import the sliced file into the 3D printer and start printed after finishing the pre-processing steps such as powder spreading. Then, put it into the ferric chloride dislocation etching solution(pH < 2) of 75 mL deionized water (Hugke Water Treatment Equipment Co., Ltd., Shenzhen, China), 25 mL hydrochloric acid (Spartan Chemical Co., Ltd., Dongguan, China), and 10 g of anhydrous ferric chloride (Aladdin Reagent Co., Ltd., Shanghai, China) for 5, 10, 15, and 20 h, respectively, then take it out using deionized water for ultrasonic cleaning 10 min, and finally put it in a DHG-9075A constant temperature blast drying oven (Yiheng Scientific Instrument Co., Ltd., Shanghai, China) at 90 °C for 2 h to obtain 3D-printed + corroded samples.

(3) Because 316L stainless steel itself has good corrosion resistance, in order to compare with the above samples, the polished samples are 316L plates bought on the market and cut into blocks of size (length × width × thickness) 10 × 10 × 4 mm, with high density and few defects.

### 2.2. Test Method and Procedure

Wettability measurement:

The wettability of the samples was measured using an OCA 20 contact angle system (Dataphysics Instruments GmbH, Filderstadt, Germany) at an ambient temperature by placing 5~7 μL drops on the surface of the sample. In order to reduce the error, each sample was measured at 5 points and averaged.

Comparison test steps of corrosion resistance of different samples:

(1) Sample preparation: choose 3D-printed, polished, 3D-printed + modified, and 3D-printed + corroded (corrosion for 15 h, standing for 15 days) samples as the test objects, and use deionized water to ultrasonically clean the samples for 10 min. Set aside after drying.

(2) Configure simulated seawater solution: take a 250 mL clean beaker, add 200 mL boiled and cooled deionized water to it, then weigh 7 g of sodium chloride (Dingshengxin Chemical Industry Co., Ltd., Tianjin, China) and use a magnetic stirrer to stir the solution until the sodium chloride is completely dissolved.

(3) Experimental measurement: the saturated calomel electrode is used as the reference electrode, the platinum sheet is used as the counter electrode, the simulated sea water solution is used as the electrolyte, and the CHI-660E electrochemical workstation (Chenhua Instrument Co., Ltd., Shanghai, China) is used to measure the open circuit potential, AC impedance and, Tafel polarization curve of the sample immersed in the electrolyte for different times.

### 3. Results and Discussion

### 3.1. Wettability Test Results

Figure 2 reflects the changes in the contact angle of water droplets on the surface of the samples etched by the corrosive solution for 5, 10, 15, and 20 h after being placed in the indoor atmosphere for different times. The sample that has been corroded for 5 h is kept in a room for 15 days, and the surface is still hydrophilic. As soon as the water drop contacts the surface of the sample, it immediately penetrates into the microstructure with

the contact angle of 0°. For the samples corroded for 10 h, after 9 days in the room, the contact angle has increased slightly. By the 15th day, the contact angle has increased to 15.3°, and the surface is still hydrophilic, unable to achieve the transition from hydrophilic to superhydrophobic state. For the samples corroded for 15 h, the contact angle of the water droplets increased greatly on the 3rd day indoors, reaching 122.4°, and on the 7th day, the contact angle of the water droplets reached 152.8°, realizing the transition from hydrophilic surface to superhydrophobic surface. For the samples corroded for 20 h, the contact angle of water droplets on the surface increased to 90.2° after being placed indoors for 5 days, realizing the transition from hydrophilic to hydrophobic. As the placement time continued to extend, by the 15th day, the surface contact angle increased to 144.2°, and the hydrophobic effect is obviously improved.

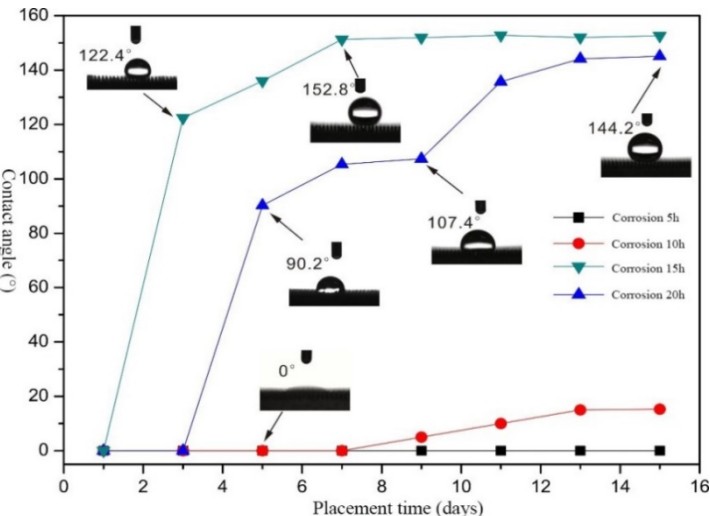

**Figure 2.** The change curve of the contact angle of 3D-printed + corroded samples with time.

This phenomenon of extreme change in the wettability of the sample surface after being placed in the air for a period of time is not an isolated case after consulting a large number of documents. P. Bizi-bandoki et al. have used laser irradiation to obtain micro–nano composite structures on metal surfaces [28–30]. These surfaces are initially hydrophilic, but after being placed in the air for a period of time, they become a superhydrophobic surface. The explanation for this phenomenon is generally believed that this time-dependent wetting state transition is mainly due to the adsorption of organic matter in the air on the surface of the microstructure, thereby reducing the surface energy. In addition, a suitable micro-nano layered structure is also a necessary condition for this transformation. For the samples prepared in this experiment, we used the field emission scanning electron microscope and X-ray photoelectron spectrometer to analyze the surface morphology and surface chemical composition of the microstructure to verify whether the cause of this experimental phenomenon is consistent with the mainstream interpretation.

Figure 3 shows the surface microstructure morphology of the 3D-printed sample etched by the etching solution for different times. Figure 3a–d correspond to the surface morphology of the sample after etching for 5 h, 10 h, 15 h, and 20 h, respectively. For the sample that has been etched for 5 h, in Figure 3a2, it can be seen that there are still some papillary structures on the sidewall of the cylinder, which have not been completely corroded by the etching solution, and as shown in Figure 3a3,a4, the surface morphology is further enlarged. It can be seen that the top surface of the cylinder is relatively smooth, and the nanostructures are not prominent. For the sample etched for 10 h, in Figure 3b2, there is no papillary structure on the side wall of the cylinder. Enlarging the image further, as shown in Figure 3b3,b4, there are faint nano-sized creases on the top of the cylinder. For the sample corroded for 15 h, from Figure 3c3, we can clearly see many *bumpy phenomena*. Further magnifying, as shown in Figure 3c4, it can be seen that these phenomena are

actually formed by nano-sized layered structures. For the sample etched for 20 h, the size of the cylinder was significantly smaller due to the long etching time. Observing the magnified images in Figure 3d3,d4 of the surface morphology, relatively light nano-layered structure can be found, which is not as obvious as in Figure 3c4.

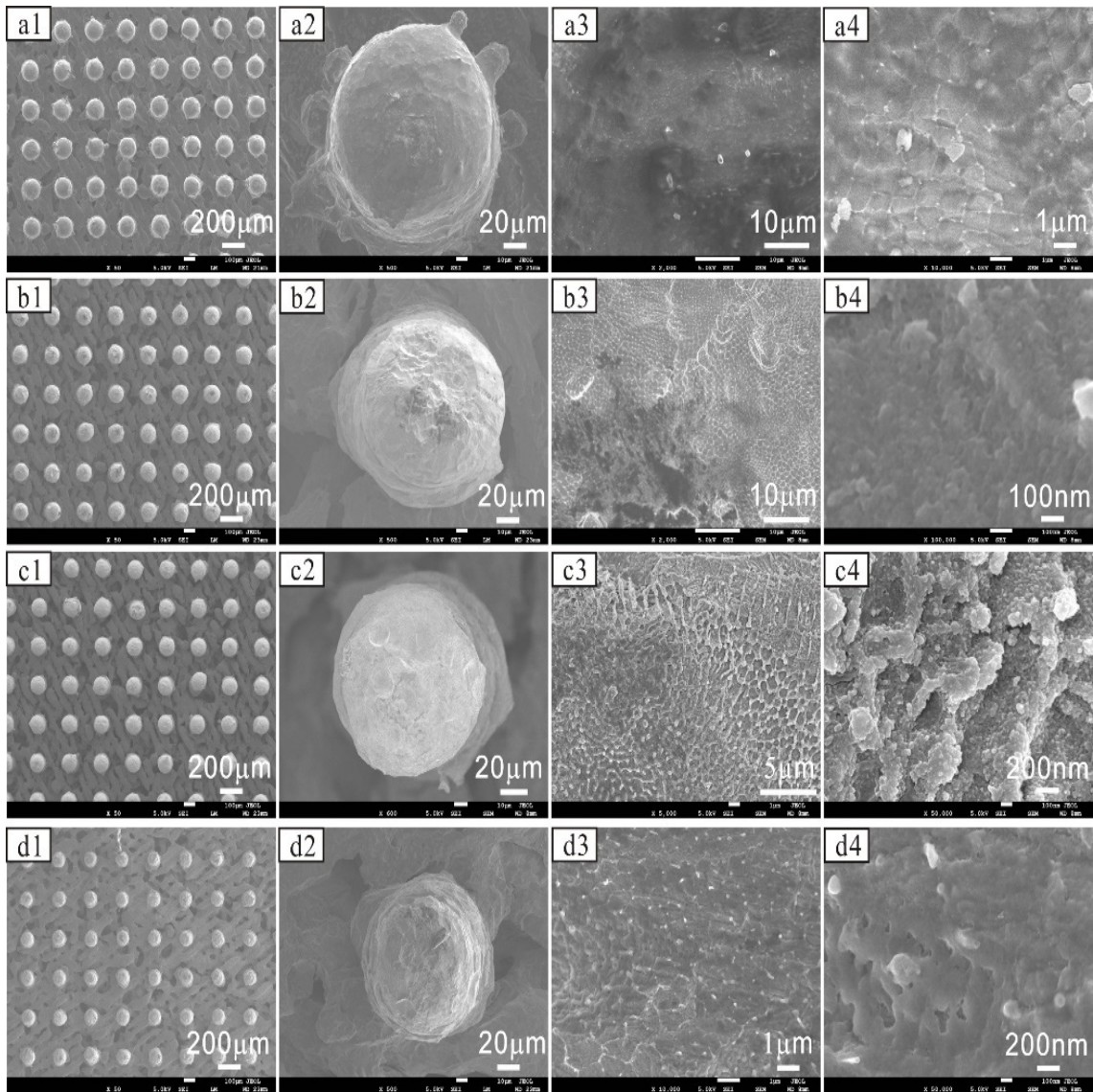

**Figure 3.** The micro-topography of the sample surface at different times of corrosion. (**a**) 5 h; (**b**) 10 h; (**c**) 15 h; (**d**) 20 h.

Through the analysis of the microstructure morphology of the sample surface, it can be found that the nano-layered structure of the cylindrical surface after 15 h of corrosion is the most prominent, and the surface is also the shortest time-consuming transition from the hydrophilic state to the super-hydrophobic state. The cylindrical surface etched for 20 h has less nano-layered structures than the one etched for 15 h, and it takes longer for the surface to change from a hydrophilic state to a super-hydrophobic state. However, the nano-layered structure on the cylindrical surface that was corroded for 5 h and 10 h was less, and the transition from hydrophilic to superhydrophobic could not be realized.

Figure 4a shows the surface XPS spectra of samples corroded for 15 h in a room atmosphere for 1 day, and Figure 4b shows the surface XPS spectra of samples corroded for 15 h in a room atmosphere for 15 days. The surface elements of the samples left for 1 day and 15 days are the same; they are Ni, Fe, O, Cr, C, and Si, all of which are 316 L

stainless steel. The difference lies in the element content. The content of the C element increased from 57.57% for 1 day to 66.45% for 15 days, and the content of the O element dropped from 38.45% for 1 day to 27.4% for 15 days. The content of the Fe element did not change significantly, and it increased by 2.17%. The increase in the content of the C element is consistent with the experimental results of P. Bizi-bandoki [28], indicating that after 15 days indoors, the microstructure adsorbs the organic matter in the air and reduces the surface energy. This is the fundamental reason why the surface of the sample changes from hydrophilic to superhydrophobic.

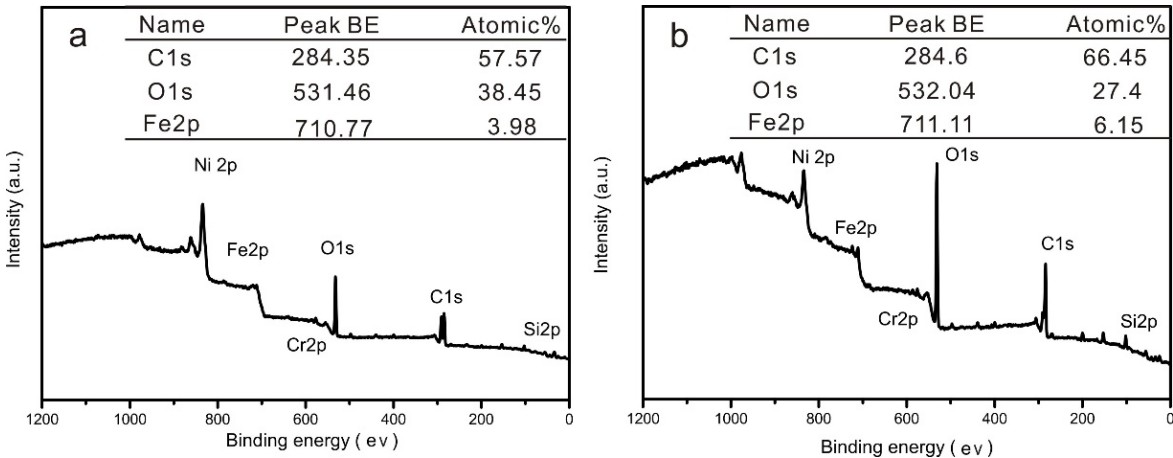

**Figure 4.** The surface XPS spectra of samples corroded for 15 h in a room atmosphere for 1 day (**a**) and 15 days (**b**).

Figure 5 shows the change in the contact angle of the water droplets on the surface of the 3D-printed + modified samples after being placed outdoors for different times. It can be seen intuitively from the figure that after two months of outdoor placement, the contact angle of the sample surface has changed from the initial 153.4° to 150.8°. It still has excellent superhydrophobic properties, indicating that a 3D-printed + modified superhydrophobic surface has good durability under natural outdoor conditions.

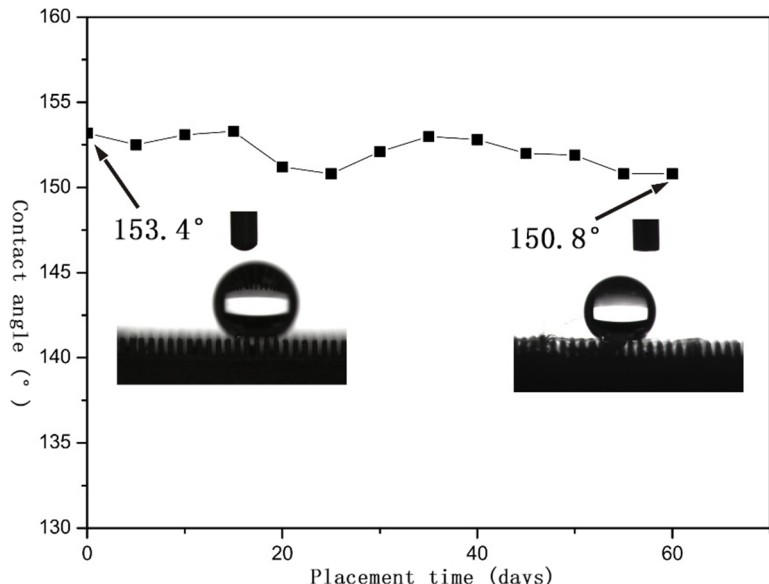

**Figure 5.** Contact angle of 3D-printed + modified samples placed outdoors for different time.

Figure 6a–c are the pictures of the contact angles of water droplets on the surface of the 3D-printed + corrosion samples (corrosion 15 h) after being placed in an outdoor natural environment for 5 days, 10 days, and 15 days, respectively. It can be seen from

the picture that on the 5th day, the contact angle of the water droplets on the surface was 151.3°, and the sample still had excellent superhydrophobic properties. However, on the 10th day, the contact angle of the water droplets on the surface dropped to 90.2°, and the superhydrophobic properties disappeared. On the 15th day, the wettability of the surface of the sample changed again, and the contact angle of the water droplets on the surface was only 29.8°, which changed from a hydrophobic surface to a hydrophilic surface. It can be found that the durability of 3D-printed + corrosion superhydrophobic samples in outdoor natural environments is less than 10 days; the durability is not good.

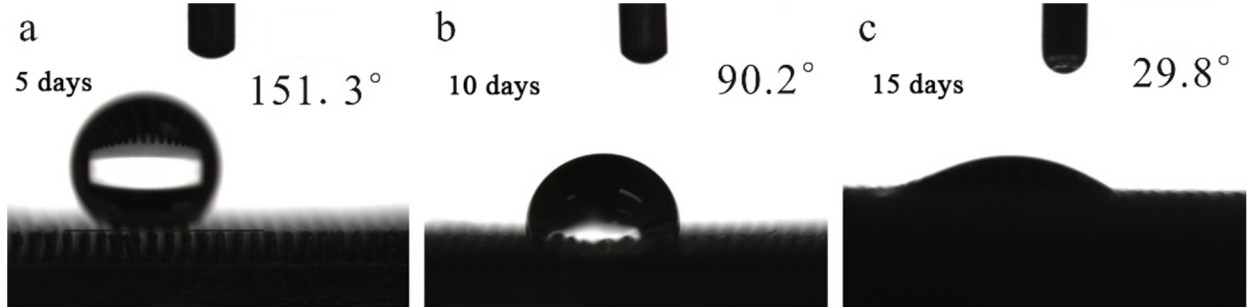

**Figure 6.** Contact angle of 3D-printed + corrosion samples placed outdoors for different times. (**a**) 5 days; (**b**) 10 days; (**c**) 15 days.

### 3.2. Comparison of Corrosion Resistance of Different Samples

Figure 7 shows the Tafel polarization curves of 3D-printed, polished, 3D-printed + corroded (corrosion for 15 h, standing for 15 days), and 3D-printed + modified samples immersed in simulated seawater for 2 h. The corrosion characteristic parameters calculated by fitting are shown in Table 1.

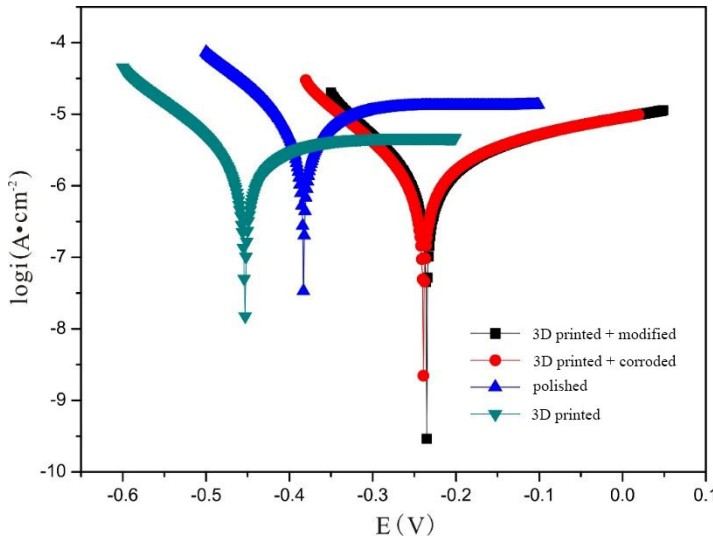

**Figure 7.** Tafel polarization curves of different samples.

**Table 1.** Corrosion characteristic parameter table of different samples.

| Parameters | 3D Printed | Polished | 3D Printed + Corroded | 3D Printed + Modified |
|---|---|---|---|---|
| $I_{corr}$ ($\mu A \cdot cm^{-2}$) | 8.953 | 2.834 | 0.34 | 0.165 |
| $E_{corr}$ (V) | −0.453 | −0.383 | −0.239 | −0.235 |

The polarization curves of the four samples in Figure 7 are arranged from right to left: 3D-printed + modified, 3D-printed + corroded, polished, and 3D-printed sam-

ples. The more to the right, the higher the corrosion potential, the greater the resistance to corrosion, and the better the corrosion resistance. Observing the specific values of corrosion current density and corrosion potential is shown in Table 1. In terms of corrosion potential, the 3D-printed sample is $-0.453$ V, the polished sample is $-0.383$ V, the 3D-printed + corroded sample is $-0.239$ V, and the 3D-printed + modified sample is $-0.235$ V; the value increases successively, indicating the corrosion resistance increases successively. In terms of corrosion current density, the 3D-printed sample is 8.953 $\mu$A·cm$^{-2}$, the polished sample is 2.834 $\mu$A·cm$^{-2}$, the 3D-printed + corroded sample is 0.34 $\mu$A·cm$^{-2}$, and the 3D-printed + modified sample is 0.165 $\mu$A·cm$^{-2}$; the value decreases successively, and the corrosion resistance increases successively. From the perspective of corrosion potential and corrosion current density, it can be concluded that the corrosion resistance of the four samples is ranked from strong to weak: the 3D-printed + modified sample > the 3D-printed + corroded sample > the polished sample > the 3D-printed sample. This is consistent with the results obtained by Sun [27], and both confirm that the surface of the chemically modified 3D printing material has good corrosion resistance.

Due to the strong polarization of the entire corrosion system during the polarization curve measurement process, it will cause greater interference to the system, bring errors to the measurement results, and lack the description of the behavior of the corrosion interface. For this reason, we continued to measure the impedance spectra of different samples.

Figure 8 shows the Nyquist curves of polished, 3D-printed, 3D-printed + corroded, and 3D-printed + modified samples immersed in simulated seawater for 2 h. It can be clearly seen from the figure that the arc radius of 3D-printed, polished, 3D-printed + corroded, and 3D-printed + modified samples gradually increases, indicating that the resistance of charge transfer is increasing, and the corrosion resistance of the sample is increasing. It can be judged that the order of corrosion resistance from strong to weak is the 3D-printed + modified sample > the 3D-printed + corroded sample > polished sample > the 3D-printed sample, which is consistent with the test results of the polarization curve.

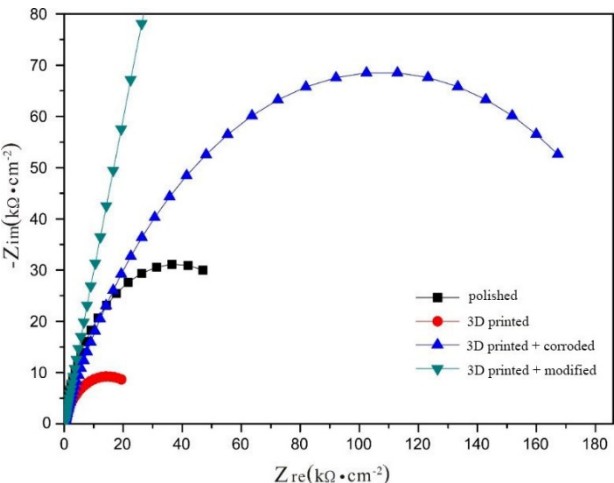

**Figure 8.** Nyquist plots of different samples.

Figure 9 is the Bode diagram of four samples. The impedance modulus at the low frequency of the Bode diagram can reflect the corrosion resistance of the material. The larger the impedance modulus value, the stronger the corrosion resistance. According to the test data, at low frequency f = 0.0178 Hz, the impedance modulus of the 3D-printed sample is $6.3 \times 10^3$ $\Omega$·cm$^2$, the impedance modulus of the polished sample is $1.27 \times 10^4$ $\Omega$·cm$^2$, the impedance modulus of the 3D-printed + corroded sample is $9.6 \times 10^4$ $\Omega$·cm$^2$, and the impedance modulus of 3D-printed + modified sample is $3.24 \times 10^5$ $\Omega$·cm$^{-2}$, and the impedance modulus becomes larger in turn. The corrosion resistance of the four samples is ranked as the 3D-printed + modified sample > the 3D-printed + corroded sample > the

polished sample > the 3D-printed sample. The analysis results of the Tafel graph, Nyquist graph and Bode graph are consistent, and the measurement results of corrosion resistance can be considered reliable.

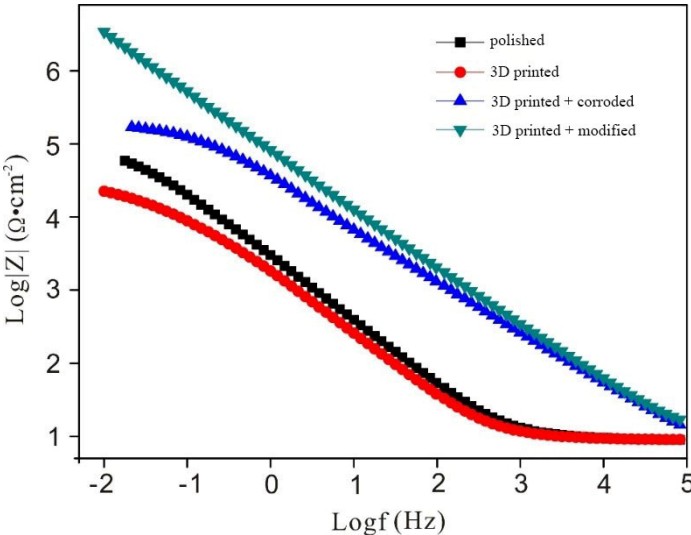

**Figure 9.** Bode diagrams of different samples.

In order to further explore the anti-corrosion mechanism of superhydrophobic surfaces and achieve a quantitative description of the charge transfer and material transfer during the corrosion process, ZSimpwin software was used to fit the measured impedance curve, and an equivalent circuit model was established with reference to relevant literature. As shown in Figure 10, for polished samples, the R (QR) model (Figure 10a) fits the measured curve best, the number of iterations Iter# is only 2 and the value of Chsq is $9.69 \times 10^{-4}$, which is also very small; Rs represents the solution resistance, CPE represents the interface capacitance between the metal and the electrolyte solution, and Rct is the charge transfer resistance. The larger the value, the greater the resistance when the charge passes through the interface between the metal and the solution. For the impedance curve of 3D-printed samples, the R(C(R(QR))) model (Figure 10b) is the best fit. The number of iterations Iter# is 3 and the value of Chsq is $3.44 \times 10^{-4}$. In this circuit model, Rp and Cp represent the resistance and capacitance of the microstructure layer, respectively. For the impedance curves of 3D-printed + corroded and 3D-printed + modified samples, the R(C(R(Q(RW)))) model (Figure 10c) is the best fit. The number of iterations Iter# is 4, The values of Chsq are $2.93 \times 10^{-3}$ and $3.20 \times 10^{-3}$, respectively. In this model, CPE represents the constant phase angle element associated with the electric double layer capacitance, and Ws is the diffusion resistance of the solution.

The element parameters in the equivalent circuit model fitted by Zsimpwin software are shown in Table 2. The charge transfer resistance Rct of 3D-printed, polished, 3D-printed + corroded, and 3D-printed + modified samples are 29,540 $\Omega \cdot cm^2$, 75,550 $\Omega \cdot cm^2$, 178,902 $\Omega \cdot cm^2$, and 207,600 $\Omega \cdot cm^2$, respectively; the value of charge transfer resistance increases successively, indicating that the corrosion resistance is successively enhanced. The value of electric double layer capacitor CPE-T can also show corrosion resistance. The smaller the value of CPE-T, the stronger the corrosion resistance. The CPE-T values of 3D-printed, polished, 3D-printed + corroded, and 3D-printed + modified samples are 89.52 $\mu F \cdot cm^2$, 67.26 $\mu F \cdot cm^2$, 20.76 $\mu F \cdot cm^2$, and 6.64 $\mu F \cdot cm^2$, respectively, and the values decreased sequentially. It also verified the correctness of the corrosion-resistance conclusion.

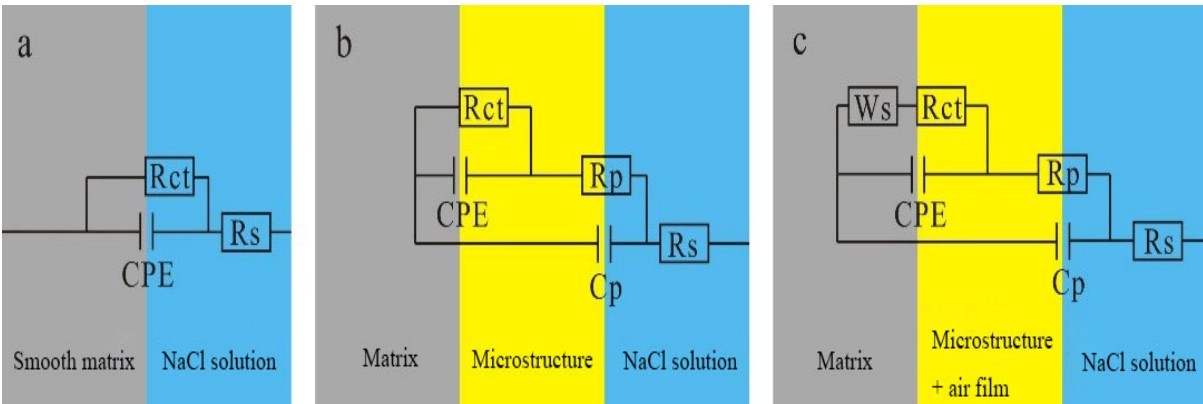

**Figure 10.** Equivalent circuit models of different samples. (**a**) Polished sample; (**b**) 3D-printed sample; (**c**) 3D-printed + corroded and 3D-printed + modified samples.

**Table 2.** Parameters of equivalent circuit components of different samples.

| Samples | Rs ($\Omega \cdot cm^2$) | Rf ($\Omega \cdot cm^2$) | Cp ($\mu F \cdot cm^2$) | Rct ($\Omega \cdot cm^2$) | CPE-T ($\mu F \cdot cm^2$) | CPE-P | Ws ($\Omega \cdot cm^2$) |
|---|---|---|---|---|---|---|---|
| 3D printed | 8.764 | 68.54 | 53.63 | 29,540 | 89.52 | 0.6 | - |
| polished | 9.364 | - | - | 75,550 | 67.26 | 0.8 | - |
| 3D printed + corroded | 9.43 | 80.22 | 0.98 | 178,902 | 20.76 | 0.6 | 0.0003 |
| 3D printed + modified | 10.51 | 83.03 | 0.22 | 207,600 | 6.64 | 0.7 | 0.0001 |

The above experimental data show that the corrosion resistance of the surface of the 3D-printed + modified sample or the 3D-printed + corroded sample is greatly improved compared with the polished sample. The main reason for this phenomenon is closely related to the surface wettability. For hydrophilic samples, the entire surface will be wetted when immersed in water, and the corrosive medium will directly contact the substrate, as shown in Figure 11a, so the corrosion reaction is more likely to happen. However, when a superhydrophobic sample is immersed in water, an air film will be formed on the surface of the superhydrophobic microstructure. As shown in Figure 11b, this film blocks the direct contact between corrosive media and the metal matrix and improves the corrosion resistance of the sample. Observing the above experimental results, we can also find that the corrosion resistance of the 3D printed sample is worse than the polished surface. The reasons for this phenomenon may be as follows: First, the surface of the sample obtained by 3D printed is hydrophilic without modification. The increase of the microstructure makes the actual solid–liquid contact area larger than the polished surface, that is, the actual contact area between the sample and the corrosive medium increases. Second, the polished sample used in the corrosion resistance test is a commercially available 316L plate. Compared with the sample obtained by laser printing, it has higher density and fewer defects, so the corrosion resistance is better.

In order to further study the corrosion resistance of the prepared superhydrophobic surface immersed in simulated seawater, the superhydrophobic samples obtained by 3D printing + modification and 3D printing + corrosion were immersed in simulated seawater solution for 1 day, 3 days, 5 days, and 7 days, then the electrochemical workstation was used to measure the changes of its Tafel curve and AC impedance curve. The measurement results are as follows:

Figure 12 shows the Tafel curve of 3D printed + modified superhydrophobic samples immersed in simulated seawater for 1 day, 3 days, 5 days, and 7 days, respectively. It can be seen from the figure that the Tafel curve does not change obviously as the immersion time increases. Combined with Table 3, the corrosion potential decreased from −0.342 V for 1 day of immersion to −0.369 V for 7 days of immersion. The magnitude of the change was small, and it was still greater than the corrosion potential of the polished surface

($-0.383$ V, see Table 1), and the corrosion current density increased from 0.319 μA·cm$^{-2}$ after immersion for 1 day to 0.743 μA·cm$^{-2}$ after immersion for 7 days, which was still lower than the corrosion current density of the polished surface (2.834 μA·cm$^{-2}$, see Table 1). Therefore, it can be concluded that the 3D-printed + modified superhydrophobic samples still have excellent corrosion resistance after immersion in simulated seawater for 7 days.

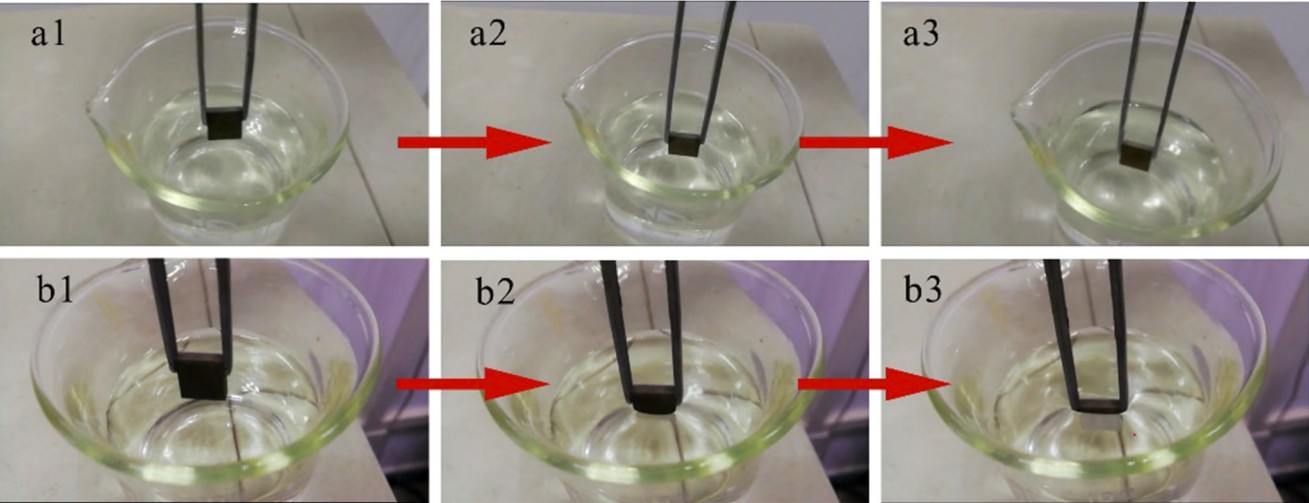

**Figure 11.** The state of hydrophilic and superhydrophobic samples when immersed in water. (**a**) Hydrophilic sample; (**b**) superhydrophobic sample.

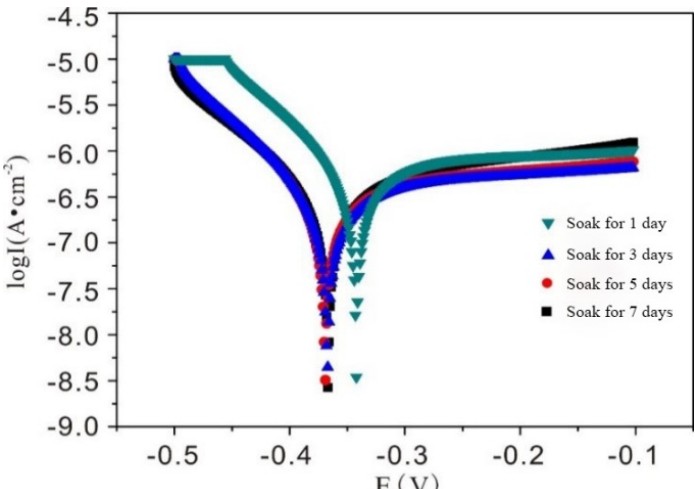

**Figure 12.** Tafel curves of 3D-printed + modified samples immersed in simulated seawater for different times.

**Table 3.** Corrosion characteristics parameter table of 3D-printed + modified samples immersed in simulated seawater for different time.

| Parameters | Soak for 1 Day | Soak for 3 Days | Soak for 5 Days | Soak for 7 Days |
|---|---|---|---|---|
| I$_{corr}$ (μA·cm$^{-2}$) | 0.319 | 0.331 | 0.387 | 0.743 |
| E$_{corr}$ (V) | $-0.342$ | $-0.367$ | $-0.368$ | $-0.369$ |

Figure 13 shows the Nyquist diagram and Bode diagram of the 3D-printed + modified samples immersed in simulated seawater for 1 day, 3 days, 5 days, and 7 days, respectively. It can be seen from the figure that as the immersion time increases, the changes in the Nyquist diagram and Bode diagram are not obvious. It also proves that the

3D-printed + modified superhydrophobic surfaces still have excellent corrosion resistance even if they are immersed in simulated seawater.

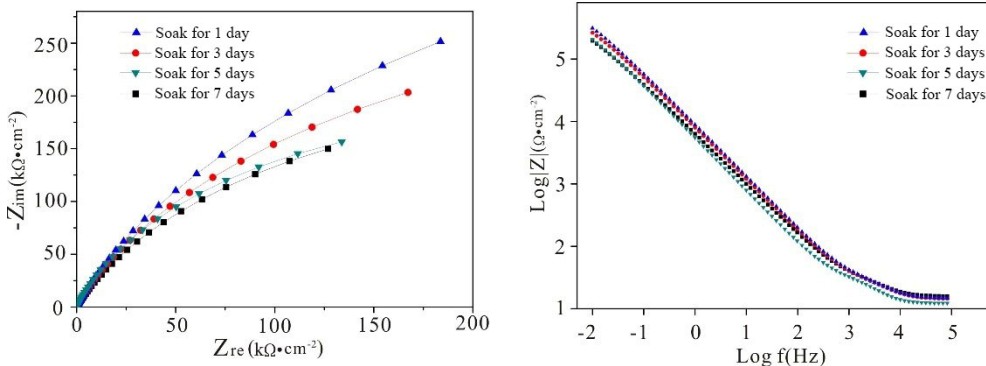

**Figure 13.** Nyquist and Bode plots of 3D-printed + modified samples immersed in simulated seawater for different times.

Figure 14 is the Tafel curve of 3D-printed + corroded samples immersed in simulated seawater for different times. From the figure, it can be seen that as the immersion time increases, the polarization curve gradually shifts to the left and the corrosion potential gradually decreases. Specific values can refer to Table 4; when the immersion time reaches 5 days, the corrosion potential drops to −0.388 V, which is lower than that of the polished surface (−0.383 V, see Table 1), and the corrosion current density increases to 3.254 $\mu$A·cm$^{-2}$, which is higher than the polished surface (2.834 $\mu$A·cm$^{-2}$, see Table 1), so the corrosion resistance of the 3D-printed + corroded sample after being immersed in simulated seawater for 5 days is worse than the polished surface, and it does not have long-term corrosion resistance.

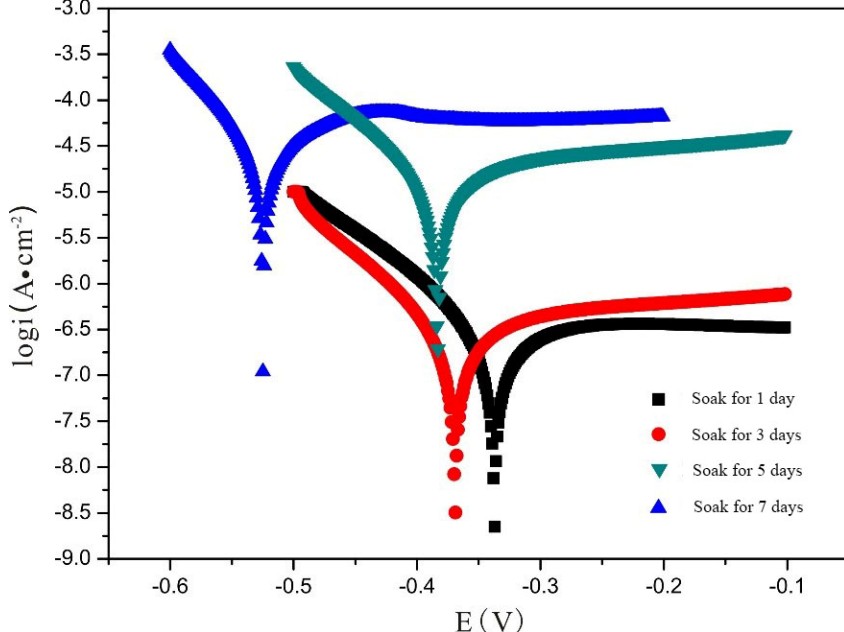

**Figure 14.** Tafel curves of 3D-printed + corroded samples immersed in simulated seawater for different times.

**Table 4.** Corrosion characteristics parameter table of 3D-printed + corroded samples immersed in simulated seawater for different times.

| Parameters | Soak for 1 Day | Soak for 3 Days | Soak for 5 Days | Soak for 7 Days |
|---|---|---|---|---|
| $I_{corr}$ ($\mu A \cdot cm^{-2}$) | 0.386 | 0.785 | 3.254 | 8.95 |
| $E_{corr}$ (V) | $-0.34$ | $-0.373$ | $-0.388$ | $-0.532$ |

Figure 15 shows the Nyquist and Bode diagrams of 3D-printed + corroded samples immersed in simulated seawater for different times. It can be found that as the immersion time increases, the radius of the arc in the Nyquist diagram becomes smaller and smaller, and the impedance at low frequencies in the Bode diagram has also been decreasing, which also shows that the superhydrophobic properties are getting worse and the samples do not have long-term corrosion resistance.

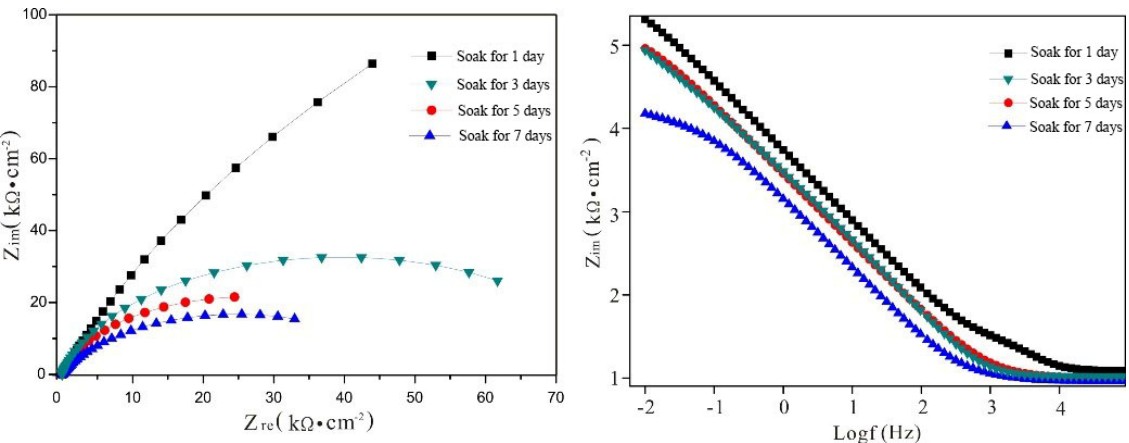

**Figure 15.** Nyquist and Bode diagrams of 3D-printed + corroded samples immersed in simulated seawater for different times.

## 4. Conclusions

In this paper, a superhydrophobic surface is prepared by Selective Laser Melting (SLM) 3D-printed technology, comparing the corrosion resistance of polished samples, unmodified 3D-printed samples, 3D-printed + modified and 3D-printed + corroded samples. The relevant experimental results are summarized as follows:

(1) For samples not treated with fluoro-silane, the microstructure adsorbs the organic matter in the air and reduces the surface energy, which is the root cause of the sample surface changing from hydrophilic to superhydrophobic.

(2) After 3D-printed + modified superhydrophobic samples are placed in outdoor natural conditions for 60 days, the contact angle of water droplets on the surface is 150.8°, which still has superhydrophobic properties and excellent natural durability; when 3D-printed + corroded superhydrophobic samples are exposed to outdoor natural conditions for 10 days, the contact angle of the surface water droplets drops to 90.2°, the superhydrophobic properties are lost, and the natural durability is not good.

(3) 3D-printed + modified and 3D-printed + corroded samples have excellent corrosion resistance when immersed in simulated seawater for a short time (2 h), and the corrosion characteristics are much higher than polished samples. In the case of prolonged exposure to simulated seawater (7 days), the corrosion resistance of the 3D-printed + modified sample has slightly decreased, and the corrosion characteristic parameters are still higher than the polished surface, showing long-term corrosion resistance; the corrosion characteristics of the 3D-printed + corroded sample has decreased significantly, and the corrosion resistance on the 5th day is already lower than that of the polished surface, and it does not have long-term corrosion resistance.

**Author Contributions:** Conceptualization, Q.Z., L.X., J.Y., Z.J.; L.X. wrote the paper; X.H. and C.D. offered useful suggestions for the preparation. All authors have read and agreed to the published version of the manuscript.

**Funding:** This research was supported by the National Natural Science Foundation of China (grant No.51875425).

**Institutional Review Board Statement:** Exclude this statement.

**Informed Consent Statement:** Exclude this statement.

**Data Availability Statement:** Exclude this statement.

**Conflicts of Interest:** The authors declare no conflict of interest.

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
