# Peer review of "Study on Preparation of Superhydrophobic Surface by Selective Laser Melting and Corrosion Resistance"

_applsci, doi:10.3390/app11167476_

Round 1

Reviewer 1 Report

Three remarks to the authors of the manuscript.

  1. Section "Introduction". At the end of the section, the aim of the work should be briefly formulated, starting with the words "The aim of the work is …".
  2. Captions to Figures 3, 8 and 9 should be expanded. It should be clear from the figure caption what is shown in the figure.
  3. Figure 1. It is recommended to add the SEM image of sample surface.

Reviewer 2 Report

This paper wants to present a new route to obtain superhydrophobic surface using Selective Laser Melting (SLM) 3D printed technology.

The article has many inaccuracies and is written in most of the text with a language too simple and sometimes not very scientific.

In particular, the induction is very weak as there is a complete lack of references to existing methods to create corrosion-resistant superhydrophobic surfaces (see ref. P.Wang et al. Corrosion science 54 (2012) 77-84, A. Benedetti et al. Colloids and surface B: Biointerfaces 137 (2016) 167–175, F. Zhang et al. Applied Surface Science 257 (2011) 2587-2591, Z. Zhang et al. Colloids and Surfaces A: Physicochem. Eng. Aspects 490 (2016) 182–188) in simulated or real marine environments but also lacks the state of the art regarding selective laser melting.

Also, the Abstract need improvements because is not clear what is the goal of the article.

Pay attention to capitalization and punctuation.

Completely missing wettability study of 3D printed + modified sample.

Statements to explain Figure 3 should be correlated by objective data such as sample roughness and pillar size after corrosion.

To support the data coming out from the corrosion studies it would be very important, if not fundamental, to have post-test observations such as wettability studies, to see if water repellency has been maintained in the samples, and SEM studies to observe if and how the samples have corroded. This is because often the corrosion resistance could be given by passivation of the surface and not of the actual resistance of the superhydrophobicity. This is considering that all Tafel curves seem to exhibit passive behaviour.

In my opinion, the article could be interesting but the way it is presented is not clear what the purpose is. A SHS has been prepared with a fluorosilane but in the text, not much importance is given to it which is however given to the corroded samples which have acquired a kind of hydrophobicity. It would be to set up the work differently.

Additional comments in the attached document and for these reasons the paper, in my opinion, is subject to major revision.

Reviewer 3 Report

The paper can be accepted after the following comments are addressed by the authors.

1. Literature should be improved. Please cite recent papers about corrosion resistant metals and some reviews about how to fabricate superhydrophobic metals (i.e. Processes 9 (4), 666, 2021)

2. The term for etching with HCl or FeCl3 is dislocation etching and not corrosion. I suggest the authors to consider the names of the samples to follow what is commonly accepted. In addition, the surfaces with the optimum performance have been prepared after combining 3D printing and etching, this is not obvious in the title of the manuscript.

3. Leaving the samples to adsorb organic matter in a not controlled environment for hydrophobization is not a reproducible method and therefore other will not be able to replicate it. I appreciate that the authors have perfomed XPS, but a more reproducible method should be used or more guidelines on how to achieve similar perfomance are required.

4. Please compare your data with state of the art examples. For example, how is the durability against corrosion for 7 days being compared with others? For such comparisons the literature update suggested above might help.

Round 2

Reviewer 2 Report

Dear Authors some corrections and annotations can be found in the attached file.

In general, the work has improved but some things would need to be fixed in particular the introduction.

In the introduction the state of the art regarding the techniques for the production of SHS is not sufficient: the two references reported (23 and 24) are two examples that do not provide an objective description of the only 2 methods reported. The drawbacks found in these two articles are not sufficient to affirm that the techniques are not valid for the production of SHS. Expand the section with other techniques widely used for the production of SH sample.

Include state of the art on SLM and SHS production. This part is critical to justify the innovation of your work. 

I appreciate the work about the wettability of the modified SHS sample. I suggest moving the text in section 3.1 reorganizing the text starting at line 425.

Last observation: why not put laser in the title? So it seems you are talking about 3D printing from the filament (as is now common use for many applications).

Reviewer 3 Report

My comments were taken into consideration and the revised version can be accepted for publication.

Author Response

Response to Reviewer 3 Comments

Point 1: My comments were taken into consideration and the revised version can be accepted for publication.

Response 1: Thank you very much for Reviewer 3’s comments, I have made further improvements to the article.